# The Effect of Lightweight Functional Aggregates on the Mitigation of Anode Degradation of Impressed Current Cathodic Protection for Reinforced Concrete

**DOI:** 10.3390/ma15051977

**Published:** 2022-03-07

**Authors:** Wenhao Guo, Jie Hu, Qijun Yu

**Affiliations:** 1School of Transportation, Civil Engineering & Architecture, Foshan University, Foshan 528225, China; w.h.guo@foxmail.com; 2School of Materials Science and Engineering, South China University of Technology, Guangzhou 510640, China; concyuq@scut.edu.cn

**Keywords:** lightweight functional aggregate, mortar, impressed current cathodic protection, acidification mitigation

## Abstract

The local acidification of secondary anode mortar was regarded as the primary reason for the degradation of the anode system, leading to a decreased service life and uneven distribution of the protection current within the impressed current cathodic protection system for reinforced concrete structures. In related previous studies, a novel type of lightweight functional aggregate was designed and prepared for the secondary anode mortar system, aiming to improve anode performance via acidification mitigation. However, the relationship between optimization effects and this functional component has not been fully clarified. In this study, two sets of experiments were carried out to investigate the effects of lightweight functional aggregates on acidification mitigation and the protection of current distribution. Research results proved that the presence of this functional aggregate was beneficial for mitigated acidification propagation and a more uniform distributed protection current, which demonstrated the importance and effectiveness of acidification inhibition on the optimization of anode performance.

## 1. Introduction

Impressed current cathodic protection (ICCP) is proved to be a very effective rehabilitation technique for corroded reinforced concrete structures under a chloride contaminated environment [1]. During ICCP treatment, the potential of the corroded reinforcement is negatively polarized to corrosion of the immune region by the external power supply; further, chloride withdrawal and alkalinity recovery [2] can be achieved as secondary beneficial effects for rehabilitation. The external anode is the most important component within the ICCP system, affecting its operating condition, protection efficiency, and service life to a great extent [3]. Therefore, the development of high performance and innovative anode materials is of great significance for further improvement of ICCP. In the past few decades, investigations on improving the performance of cementitious anode materials were mainly focused on the optimization of its conductivity with conductive enhancement materials, e.g., carbon fibers [4,5], graphite [6], etc. These components generally exhibit low resistivity (10-2~10-3 Ω·cm for carbon fibers [7] and 10-3 Ω·cm for graphite [8]) and are able to decrease the bulk resistivity of cementitious anode materials for one or two orders of magnitude [9]. However, the degradation and premature failure of the ICCP system is basically related to the local acidification damage in the external anode system [10,11,12]. During ICCP treatment, a series of electrochemical reactions occur on the surface of the primary anode [13]:2OH^−^ → 1/2O_2_ + 2e^−^ + H_2_O(1)
H_2_O → 1/2O_2_ + 2e^−^ + 2H^+^(2)
2Cl^−^ → Cl_2_ + 2e^−^(3)
Cl_2_ + H_2_O → Cl^−^ + ClO^−^ + 2H^+^(4)
C + 2HOCl → CO_2_ + 2HCl(5)

The above reactions affect the dissolution equilibrium between different hydration products and ions in the cement pore solution, causing the dissolution of hydration products (e.g., Ca(OH)_2_, C-S-H gel, etc.) [14] and subsequently, increased porosity [15] and decreased mechanical properties [16] of the external anode mortar in the vicinity of the primary anode [10,17]. Although the above-mentioned conductive modification could decrease the IR drop of the whole external anode system (mainly from the electronic conduction perspective [18]) and improve its performance, it contributes little to the mitigation of the acidification degradation process. For example, Bertolini et al. [17] discovered that the anode system with carbon fiber reinforced mortar could only be used at 50~100 mA/m^2^ (relative to the primary anode metal) for about 3000 A-h/m^2^ in total charge density. The expected anode performance and service life were far from what can be expected in the chloride-contaminated alkaline solution as specified in NACE TM 0294-2007 (total charge density above 38,500 A-h/m^2^ under a current density of 8900 mA/m^2^ [19]). The insufficient resistance to local acidification is believed to be the primary reason behind this huge gap. Therefore, superior resistance to anodic acidification damage should be seriously considered for the design of a high-performance external anode within the ICCP system.

In addition, the ions migration during the ICCP treatment exhibited a strong influence on the operating parameters and acidification damage in the external anode system [20]. Despite the qualitative simulation that was carried out in the previous studies, the mechanisms that determine the durability of the external anode system still remain unclear: based on the Nernst–Planck equation for ion diffusion, the calculated erosion range was within 100 μm of the primary anode surface after 10 years of operation [11], which is several times different from what has been observed in experimental results [16,21]. Moreover, it was found that the characteristics of the anode affected the concentration profile of the chloride during the electrochemical treatment: the accumulation of extracted Cl^−^ in the anode mortar would result in a higher re-diffusion risk and further affect the protection efficiency, therefore, a much thinner anode mortar (65% thinner compared with controlled thickness) was suggested to decrease the amount of extracted substance so as to mitigate the side effect [12,22], despite knowing that this scheme was not conducive to the stability of the anode system.

In addition, the investigations on advanced anode materials were mainly carried out with isolated anode samples in the reported studies: in most cases, researchers paid more attention to the resistivity reduction in the anode mortar with carbon fibers [5,23], carbon black, or carbon nanotubes [24]. Despite the conductive enhancing effects and the resulted optimized anode performance discussed, the relationship between the anode mortar and the overall performance was ignored.

In our previous study, a novel type of lightweight functional aggregate (LFA) with low bulk resistivity and excellent acidification resistance was designed and prepared, and better polarizing stability and extended service life of the anode system were obtained [25]. The feasibility of enhancing the anode performance by acidification mitigation was proved, and this technical route has been tracked by relevant scholars [26]. However, the influence of the high-performance anode mortar based on the LFA on the protection efficiency of the ICCP system was not discussed. In this paper, the simulated ICCP system was designed and prepared based on actual operation parameters; the pH of the simulated pore solution in the anode cell and the anode potential were periodically monitored to establish a correlation between the anode acidification process and the anode performance. The relationship between the acidification mitigation effect of the LFA and the protection efficiency of the ICCP system was discussed in detail based on the derived results.

## 2. Experimental

### 2.1. Materials

In this study, ordinary Portland cement (P II 42.5, China Resources Cement Holdings Ltd., Guangzhou, China), together with ground-granulated blast furnace slag (grade S95, Hualisheng Concrete Co., Ltd., Huizhou, China), secondary fly ash (Class C, Hualisheng concrete Co., Ltd., Huizhou, China) in accordance with relevant requirements in GB/T 1596–2017, and silica sand with a fineness of 2.7 were applied as cementitious materials. The detailed chemical compositions of these raw materials are presented in Table 1. Crushed limestone with a particle size range of 5–31.5 mm was applied as the aggregate for the preparation of a concrete bridge in the simulated ICCP system.

Tap water was used to prepare the anode mortar specimens and the concrete bridge. Deionized water was used for the preparation of the simulated concrete pore solution. Polyamide (PA-66) was applied as a base insulating and corrosion-resistant material for the preparation of the simulated ICCP system cell case. A mixed-metal oxide (MMO)-coated titanium wire (Zhongtai Metal Materials Co., Ltd., Shanxi, China) with a diameter of 2.00 mm and Q235 low carbon construction steel (Shaoguan Iron and Steel Plant, Shaoguan, China) with a diameter of 8.00 mm were applied as the primary anode and cathode, respectively, in the simulated ICCP system. Other raw materials (e.g., agar powder, porous ceramsites, carbon fiber, and analytical grade chemical reagents) can be found in our previous study [27,28].

### 2.2. Experimental Setup

This study consists of two major sets of experiments:(1)The long-term monitoring based on simulated ICCP system: this experiment was designed to investigate the influence of LFA on the chloride migration behavior and the relationship between anode acidification and anode performance during ICCP treatment.(2)The potentiostatic polarization test of the anode specimens: this experiment was designed to characterize the influence of the LFA on the acidification accumulation and current distribution uniformity of the external anode mortar.

#### 2.2.1. Simulated ICCP System

The simulated ICCP system in this study consisted of two electrolytic cells and a concrete bridge, as shown in Figure 1. The mixture design of the concrete bridge is presented in Table 2. After standard curing for 6 months, the concrete bridge (with dimensions of 40 × 30 × 30 mm) was cut from the concrete specimen (with dimensions of 100 × 100 × 100 mm) and then fixed between the two electrolytic cells as shown in Figure 1. Two surfaces of the concrete bridge were exposed to the electrolytic cells and the other surfaces were sealed with epoxy resin. A 40 mL chloride-contaminated (3.5 wt% NaCl) simulated concrete pore solution (the chemical compositions of the simulated concrete pore solution were consistent with our previous study [25]) was poured into the electrolytic cells. After that, the cathode cells were filled with the same volume (26 cm^3^) of siliceous river sand, while the anode cells were filled with mixed aggregates; specific mix proportions of the aggregates in the anode cells are shown in Table 3. MMO-coated anode Ti wire and Q235 construction steel were then separately immersed in the anode and cathode cells. The area ratio of the anode electrode, cathode electrode, and exposed concrete bridge surface was 1:19:33, which was similar to that of the ICCP system in the relevant reported study [17]. After that, the anode and cathode electrodes were connected to the power supply, and the cathodic polarization of 200 mA/m^2^ was applied to Q235 construction steel, equaled to the anodic current density of 6700 mA/m^2^ for the MMO-coated anode Ti wire.

#### 2.2.2. Experimental Setup for Acidification Accumulation and Conductivity Distribution

The experimental setup used to characterize the influence of the LFA on the acidification accumulation and conductivity distribution of the external anode mortar was described in detail in our previous study [28]. The geometric dimension of the cylindrical anode mortar sample was 30 mm in height and 30 mm in diameter. The MMO-coated Ti wire embedded in the anode mortar was 60 mm in length. After casting, the specimen was cured at 20 ± 2 °C and relative humidity of 95% for 28 days. After curing, the top and bottom surfaces of the cylindrical sample were sealed with epoxy resin. Two major categories of anode samples were prepared in this test. The neat paste samples were prepared to characterize the influence of the hydrated matrix on the propagation of local acidification, and the mortar samples were prepared to characterize the influences of aggregate type and volume on the propagation of acidification. The specific information of the anode mortar sample is shown in Table 4.

### 2.3. Testing Methods

#### 2.3.1. Characterization of the Influence of the LFA on the Chloride Migration Behavior during ICCP Protection

The polarization potential (vs. saturated calomel electrode (SCE)) of the MMO-coated Ti wire of the simulated ICCP protection system was periodically monitored. Meanwhile, the acidification process in the anode cell was monitored by measuring the pH of the simulated concrete pore solution in the anode cell by Mettler Toledo FE20K pH meter (Mettler-Toledo Ltd., Leicester, UK). The influence of the aggregate type and volume on the chloride migration process during the ICCP treatment was quantitatively determined by ion chromatography (IC) with Dionex ICS-900 (chromatographic column capacity ≥ 180 μmol, flow accuracy ≤ 0.1%), Thermo Fisher Scientific (China), Co., Ltd., Shanghai, China. About 1 mL of simulated concrete pore solution was periodically collected from both the anode and cathode cells for IC measurements, and the same volume of fresh prepared simulated concrete pore solution was re-filled into each cell to compensate for the loss of sampling. Particularly, the pH of the re-filled solution was adjusted to the same pH of the simulated concrete solution in the electrolytic cells by using nitric acid to ensure that this sampling did not interfere in the acidification process.

The influences of anodic acidification on the morphology of MMO-coated Ti wire was observed using scanning electron microscopy (SEM, ZEISS EVO 18, Carl Zeiss NTS GmbH., Oberkochen, Germany). The accelerating voltage of the beam was 10 kV and the magnification of the SEM imaging was 500×.

#### 2.3.2. Test Methods for the Potentiostatic Polarization Test

The experimental setup of the potentiostatic polarization test was consistent with a published contribution [26]. Different from the constant loop current mode in the accelerated acidification test (AAT), the anode polarization potential was controlled constant at 1200 mV vs. SCE. Since the polarization potential was controlled, the loop current was recorded daily with a potentiostat.

After this test, the anode specimens were sampled to observe the acidification status. The preparation methods of fluorescent resin-impregnated thin sections and instrument parameters for the fluorescence microscope (ZEISS Discovery V12, Carl Zeiss NTS GmbH., Oberkochen, Germany) observation were in line with our previous study [27].

#### 2.3.3. Calculation Methods of Equivalent Diffusion Coefficient

The obtained anode current in potentiostatic polarization test was closely related to the mass transfer process of hydroxyl ions. During which, the pore structure of the anode mortar was the leading influencing factor and the reaction that occurred on the primary anode surface was believed to be a “diffusion controlled” process. Therefore, the relationship between the anode current and the reactant diffusion behavior can be described with a Cottrell equation [28]:(6)$$i=\frac{{\mathrm{nFAC}}_{j}^{0}\sqrt{D_{j}}}{\sqrt{\pi t}}$$
where i (A) is the current, n is the number of electrons, F is the Faraday constant (C/mol), A (cm^2^) is the area of the primary anode, Cj^0^ (mol/cm^3^) is the initial concentration of reaction substance J (mainly hydroxyl ion in this case), Dj (cm^2^/s) is the diffusion coefficient of the reaction substance J, and t (s) is the reaction time.

Since the reaction, in this case, occurred on the surface of the MMO-coated Ti wire, it can be treated as a tubular band electrode (TBE). The diffusion-controlled process can be described as follows [29]:(7)$$\frac{1}{D_{j}}\frac{\partial C}{\partial t}=\frac{\partial^{2}C}{\partial R^{2}}+\frac{1}{R}\frac{\partial C}{\partial R}+\frac{\partial^{2}C}{\partial Z^{2}}$$
in which:t > 0, 0 ≤ R < R_0_, −∞ < Z < ∞(8)
particularly, when
t = 0, 0 ≤ R ≤ R_0_, −∞< Z < ∞, C = Cj0(9)
when
t > 0, R = R_0_, |Z| ≤ W/2, C = 0(10)
For other non-reactive surfaces, when
(11)$$R\;=\;R_{0},\;|Z|\;\geq \;W/2,\;\frac{\partial C}{\partial R}=0$$
where R_0_ is the diameter of TBE and W is the length of TBE. Z and R are the spatial coordinates of the distance from the electrode surface at time t. C is the concentration of the reactant. Despite the complexity of the anode mortar and the oxidation of the hydroxyl ion is not the only anodic reaction that occurred on the surface of the primary anode, the exact value of the diffusion coefficient of the reactant cannot be obtained with the Cottrell equation and related derivation. However, based on the above analysis, a semi-quantitative relationship between the mass transfer capability of the anode mortar and the anode current can still be obtained as follows:(12)$$D_{A}\propto i^{2}t$$
where D_A_ can be summarized as an equivalent diffusion coefficient to characterize the diffusion behaviors of the anode mortar, and the time-dependent alteration of D_A_ can be related to the structural alterations that resulted from the local acidification.

#### 2.3.4. Preparation and Test Methods for the Accelerated Acidification Test (AAT) and Anode Conductivity Uniformity Test

The preparation procedures of the samples applied for this test were the same as what has been stated in Section 2.2.2. Based on our previous preliminary research results, the anode mortars with 0.75 vol% of carbon fiber and 40 vol% of LFA (LFA-40+cf), 0.75 vol% of carbon fiber and 40 vol% of NA (NA-40+cf), and 40 vol% of LFA (LFA-40) and 40 vol% of NA (NA-40) were selected to maximize the influences of aggregate type on the conductivity uniformity for both systems.

The conductivity uniformity test was executed along with the continuous extension of AAT. During the AAT (constant current mode), the suffered current density was 200 mA/m^2^ for the active mortar surface and 3000 mA/m^2^ for the embedded primary anode surface; the setup for AAT was correspond with [27]. Specifically, the conductivity uniformity test was executed at 0 d (initial state), 7 d, 14 d, 21 d, 28 d, and 42 d of AAT, respectively.

A 4-electrodes 3-circuits testing system was applied in this test. As presented in Figure 2, the 4-electrodes refer to the embedded primary anode wire as the polarizing electrode in the center and the other 3 current receiving probes that are evenly and symmetrically distributed in the horizontal plane. The specific contact sites of the receiving probe were identified by the marks at the upper surface in advance. Five scanning planes were set for each anode sample and each test surface was provided with 9 testing points.

During the conductivity uniformity test, the current was imposed from the primary anode, then traveled through the anode mortar and eventually picked up by each receiving probe. Therefore, the only variable in this system was the resistance of the secondary anode mortar. The voltage between the primary anode wire and receiving probes was 3 V.

## 3. Results and Discussion

### 3.1. The Simulated ICCP System

#### 3.1.1. The pH Value of Simulated Concrete Pore Solution in Anode Cells

In general, owing to the electrochemical reactions that occurred on the anode site, the hydroxyl ions would be consumed. As the acidification continues, because of the solubility equilibrium between alkaline hydration products and ions in pore solution, hydrated products would be dissolved as a result. According to related contributions, Ca(OH)_2_ was the first hydration product to undergo dissolution [30]. Therefore, the equilibrium pH of Ca(OH)_2_ (12.65) can be regarded as the critical pH at which the dissolution of the hydration products begins. As presented in Figure 3, the measured pH value of all anode cells presented a clear downward trend: the pH of reference anode cells (1-1,2,3) decreased sharply after polarizing for about 600 to 700 C of accumulative electric charge quantity. However, the anode cells with the LFA generally present a delayed time node of declining pH: the pH value of anode cells with 15 vol%, 25 vol%, and 45 vol% of LFA decreased to lower than 12.56 after polarizing for 1000 C, 1200 C, and 2000 C, respectively. The obtained results indicated that the presence of the LFA could effectively respond to the acidification, and the higher the volume of the LFA was, the more significant prolonging effect on acidification inhibition could be expected: taking samples (1-1) and (4-3) as an example, when the anode pH value fell below 12.65, the accumulative electric charge quantity was 2005 C for sample (4-3), 224% longer than that of sample (1-1), which is equivalent to 1275 days of service life extension for a primary anode that polarizes with a current density of 200 mA/m^2^.

#### 3.1.2. The Anode on Potential and Surface Morphology

The maintenance of the anode alkaline environment-induced alterations in the monitored anode potential. As presented in Figure 4, the anode potential of different simulated ICCP testing samples varies within the monitoring period, the average value of the potential anode was about 900 mV vs. SCE in the early stage but increased to around 1200 mV vs. SCE at the end of this test. It is noteworthy that the mutation of the anode potential is highly coincident with the time node when significant acidification occurred.

Taking sample (1-1) as an example, when the accumulative electric charge quantity was 730 C, the anode potential was 822 mV vs. SCE, while the potential elevated to 1010 mV vs. SCE after being energized with another 70 C. The upward trend of the anode potential was believed to be related to the degraded reaction environment: as the hydroxyl was consumed continuously within the anode cell, a higher polarizing potential was needed and a greater potential difference occurred as a result. The potential difference of the anode cells with NA was around 400 mV, while that of the anode cells with 40 vol% of LFA was only about 200 mV throughout the entire testing period. Generally, a relative noble polarizing potential was believed to be beneficial for the operation of the ICCP system.

For the relative higher potential of samples with the LFA compared with those with NA within the early stage of the ICCP test (0~800 C), the adsorption of chloride ions by the LFA was considered to be a possible explanation; related evidence to support this hypothesis is discussed later in the IC test results.

Figure 5 illustrates the surface morphology of different MMO wires. Figure 5a presents the original surface morphology of an MMO wire, the microcrack on the dark black colored surface was induced by the sintering of the MMO on the titanium substrate [31], and the raised granular-like particles were the precipitated IrO_2_ crystal [32]. After being energized with over 3200 C, several structural alterations occurred on the surface of the MMO wire, as presented in Figure 5b. The original dark-black-colored MMO-coating decomposed severely, exposing the white-colored substrate and the generated titanium oxides. Based on a relevant study, such alterations can be expected on an anode that suffered from long-term acidification under high current density [33], and the degradation of the sintered MMO coating integrity would reduce the catalytic efficiency severely [34]. By comparison, the MMO-coating of anodes that immersed in cells with LFAs (Figure 5c–e) were relatively intact, and fewer oxidation products can be observed as well. The modification effects were positively correlated with the volume of the LFA and the alkaline environment maintaining effects as discussed in Figure 3 were believed to be related to a better reaction environment (high alkalinity). The above results indicate that the erosion damage on the primary anode can be mitigated with an environment in which alkalinity can be maintained, and the addition of the LFA in the secondary anode system is a feasible technical route to realize this purpose.

In summary, the accumulation of anodic acidification effects at the anode site acidized the pore solution, degrading the hydration products. The presence of the LFA was beneficial to the mitigation of the anode acidification and furthered the operational stability of the ICCP system.

#### 3.1.3. The Chloride Profile in Simulated Concrete Pore Solution

The tested chloride concentration of the sampled pore solution in different operation stages of the ICCP treatment was presented in Figure 6. It was observed that the chloride concentration in all cathode cells decreased while those of the anode cells increased along with the treatment time, known as “chloride withdrawal effect” [35]. Furthermore, the variations in chloride concentrations can be related to the differences in anode aggregate composition: the cathode cell of sample (2-1) was tested to have the lowest chloride concentration (0.552 mol/L) after being energized with only 760 C, and that of sample (2-1) and (1-1) were lowest compared with the other cathode cells after being energized with 3200 C. For the anode side, the tested results were not quite in line with expectations: normally, the chloride concentration of an anode cell would be higher than the initial value. However, the chloride concentration of samples with LFA were much lower than the initial contaminating value after being energized with 760 C: the mean chloride concentration of anode cell contained 15 vol%, 25 vol%, and 40 vol% of LFA was 0.61 mol/L, 0.56 mol/L, and 0.51 mol/L respectively, which is much lower than the reference cell and even the original contaminated value. Similar results were maintained in the later operating stage. Since the reduction in chloride concentration is closely related to the volume of the LFA, the most possible explanation for this phenomenon is the chloride ions were absorbed by the modified agar gel that impregnated the LFA.

In addition, the chloride adsorption effects induced by the LFA could also explain the relative high polarizing potential of the anode with LFAs at the early stage of the ICCP testing period. Since the presence of chloride ions in the pore solution was beneficial for the anodic reaction (1-3) and (1-4), the adsorption effect of the LFA was responsible for the increased potential at the early stage.

### 3.2. Potentiostatic Polarization Test of Anode Specimens

#### 3.2.1. The Alteration of Diffusion Behavior of the Anode Mortar

The calculated equivalent diffusion coefficients of the anode samples prepared with neat cement paste are presented in Figure 7.

Firstly, based on the above analysis, the higher equivalent diffusion coefficient of the anode sample was believed to be a result of the loose pore structure of the anode mortar: the sample with a higher W/C ratio corresponds to a higher initial equivalent diffusion coefficient. In addition, the calculated equivalent diffusion coefficients were positively correlated with the polarizing time. The longer the polarization period, the higher the diffusion coefficient in the anode mortar; the relationship between the two parameters was almost linear.

Specifically, an anode with a higher W/C ratio corresponded to a higher increasing trend in the calculated equivalent diffusion coefficient: the fitted slope of the sample (W/C = 0.5) was 1.46 × 10^−7^ cm^2^/(s·h), which is 205% higher than that of the sample (W/C = 0.3). In related publications, a lower W/C ratio is usually equipped with a lower erosion extension rate [36]. This result was in line with relevant research contributions.

Figure 8 presents the fluorescent impregnated cross section of anode samples after 280 h of polarization. It can be clearly observed that the sample with a lower W/C ratio (0.3) corresponds to a slighter fluorescence effect in the anode mortar portion. By contrast, the sample with a higher W/C ratio (0.5) presented a sharp aperture with a diameter of over 3000 μm, indicating a more severe acidification propagation. The observed results were in agreement with the previously mentioned calculation results.

Figure 9 presents the calculated equivalent diffusion coefficient of anode samples with varied carbon fiber content, aggregate type, and volume fraction. Based on the former analysis, the slope of the linear fitting of the equivalent diffusion coefficient can be applied to characterize the anode acidification extension rate. Moreover, the fitted residual sum of squares (RSS) can be used as another parameter to characterize the stability of each sample during the testing period. The fitted slope and RSS are presented in Table 5.

According to the statistics, sample NA-20, NA-30, NA-40, and LFA-20 were the ones with a high-level increasing trend and volatility. The fitted results were even higher than that of neat cement paste with the same W/C ratio (0.86 × 10^−5^ cm^2^/(s·h)), which indicates that the introduction of the aggregate phase or the low volume fraction of the LFA phase would accelerate the acidification degradation of the anode system. In general, samples with a high-volume fraction of the LFA phase (higher than 30 vol%) present lower fitted slopes and slighter numerical fluctuation. These results demonstrated that the presence of the LFA within the anode system has a positive influence on the maintenance of anode performance.

To make a better comparison of the degradation effects, a fluorescence micrograph of sample NA-20 (Figure 10a) and sample LFA-40+cf (Figure 10b) were selected. Compared with (Figure 10b), a much brighter “erosion ring” can be observed in Figure 10a, indicating more severe erosion damage in the vicinity of the primary anode. Related observation results also proved the reliability of the above calculation results and characterization methods.

#### 3.2.2. The Alteration of Anode Conductivity Uniformity

Figure 11, Figure 12, Figure 13 and Figure 14 present the monitored current distribution of anode samples during the AAT.

The nine coordinates in the figure correspond to the nine testing points of each horizontal cross-sectional plane. The gray dotted inner and outer circles indicate 0 % or 200 % of the sample’s mean received current. Similarly, the orange dotted circle indicates the sample’s mean received current.

For all selected samples, the measured five circles were all relatively close to the mean received current at their original state, and exhibited satisfactory uniformity in conductivity. However, with the extension of AAT, the received current fluctuated in varying degrees: the received current of the anode with carbon fibers tended to present a higher fluctuation amplitude, especially in later monitoring periods (28 d and 42 d). Taking NA-40+cf as an example, after 42 days of AAT treatment, the current received at detecting point-3 and point-7, layer-4, were over 100% higher than the average current value. By contrast, the anode with the LFA generally presented a relatively mild fluctuation at the later testing periods in the anode system with or without carbon fibers.

To quantitatively characterize the alterations of conductivity uniformity, the normalized standard deviations of tested data of each sample in each monitoring period were calculated. As presented in Figure 15, for sample NA-40, the standard deviation increased sharply with the extension of AAT treatment: before AAT, the standard deviation was 0.185, but it increased to 0.324 and 0.371 after 7 days and 14 days of AAT. By contrast, the standard deviation of LFA-40 was 61% lower at the same time. In addition, it is worth noting that samples with carbon fibers present a relatively higher increasing trend. This phenomenon can be explained by the degradation of the conductive network that formed by admixed carbon fibers. According to a related publication, the porosity of hydrated cement matrix was the main influencing factor in the connectivity between carbon fibers [18,37]. Therefore, the acidification would increase the porosity of the cement matrix and the form of erosion was inhomogeneous; as a result, the heterogeneity of the conductive path increased.

In summary, the presence of the LFA in the anode mortar system was beneficial for the acquisition of a more uniform current distribution, and this conclusion was proved in the anode mortar system with or without carbon fibers.

## 4. Conclusions

In this study, the effects of the LFA on the global performance of the ICCP system, especially the chloride withdraw effects, were investigated based on the designed simulated ICCP testing apparatus. Particularly, the influences of aggregate type and carbon fiber on the conductivity uniformity were characterized based on the AAT-combined conductivity uniformity tests scheme. The obtained test results emphasized the importance and revealed the related beneficial effects of anode acidification inhibition within the ICCP system. The main conclusions in this study can be summarized as follows:

The presence of the LFA in the anode mortar within the ICCP system could effectively mitigate the anodic acidification, calculated with the simulated concrete pH lower than 12.65; the onset of severe anodic acidification was relatively delayed by 63%, 84%, and 177% of the reference samples with the increasing displacement of the LFA volume.

Based on the test results from the simulated ICCP testing apparatus, the presence of the LFA was beneficial for a decreased chloride concentration in the anode cells, which was beneficial for a reduced re-migration risk after the ICCP treatment. The potential adsorption capability of the impregnated modified agar gel within the LFA could be the most reasonable explanation for the decreased chloride concentration.

Compared with the conventional aggregate type (NA), the presence of the LFA was beneficial for a decreased propagation rate of acidification erosion within the anode mortar system. The growth rate of the equivalent diffusion coefficient of the anode sample with 40 vol% of the LFA was 43% of that of the sample with the same volume of NA.

The anodic acidification resulted in orientated erosion at the anode mortar system and further decreased the anode conductivity uniformity, especially in those with carbon fibers and conventional aggregates. The presence of the LFA could retard the increase in conductivity heterogeneity and thus is beneficial for the acquisition of a more uniform distributed protection current.

## 5. Patents

Patents relevant to this article include: China national invention patent ZL201510527591.2, ZL201710812074.9, and PCT/CN2017/118425.

## Figures and Tables

**Figure 1 materials-15-01977-f001:**
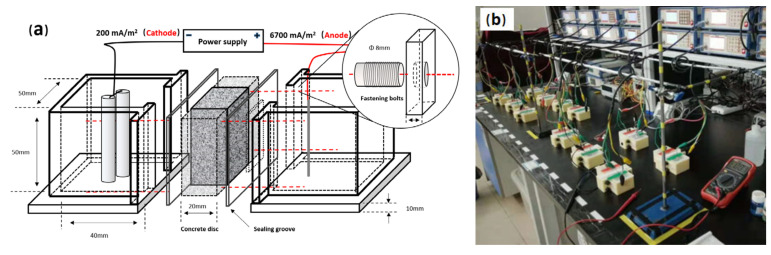
The schematic diagram of the electrolytic tank and concrete disc (**a**) and an experimental photo of the assembled testing devices (**b**).

**Figure 2 materials-15-01977-f002:**
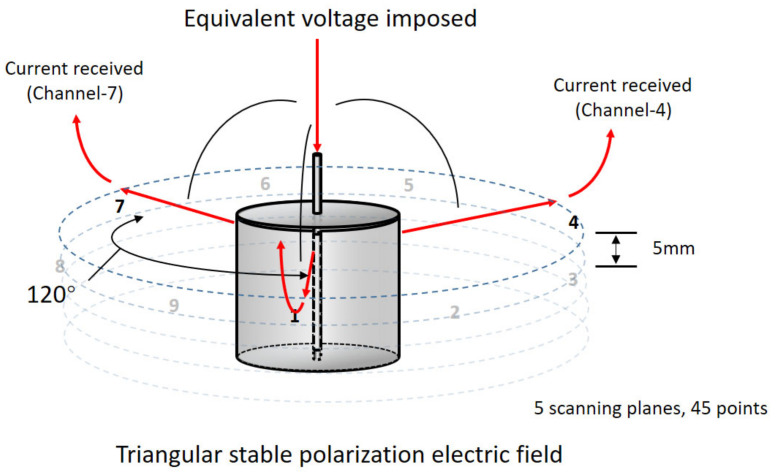
The schematic diagram of the conductivity uniformity test for anode mortar.

**Figure 3 materials-15-01977-f003:**
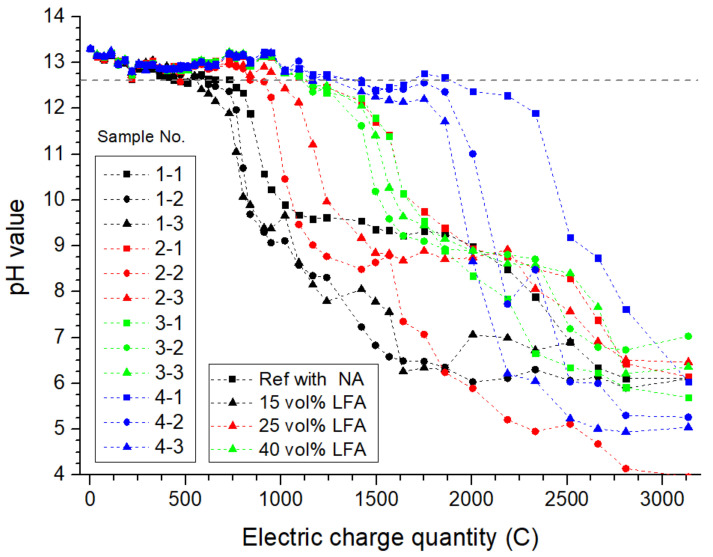
The monitored pH values of simulated concrete pore solution in all anode cells.

**Figure 4 materials-15-01977-f004:**
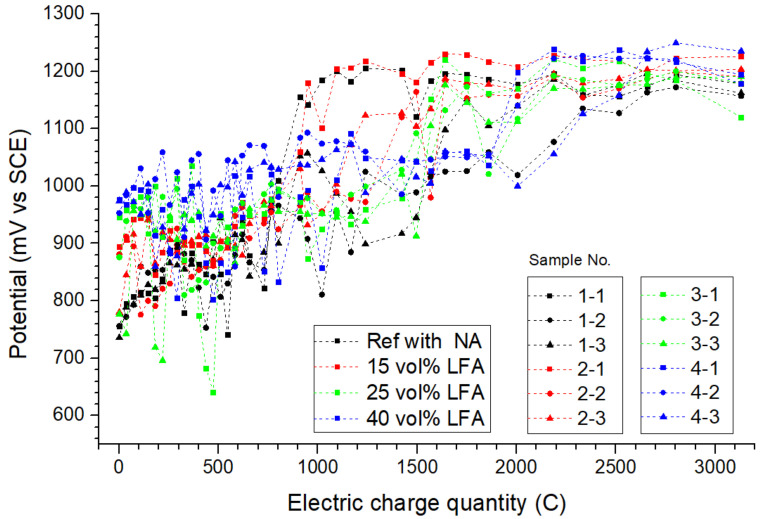
The monitored anode on potential in all simulated ICCP testing samples.

**Figure 5 materials-15-01977-f005:**
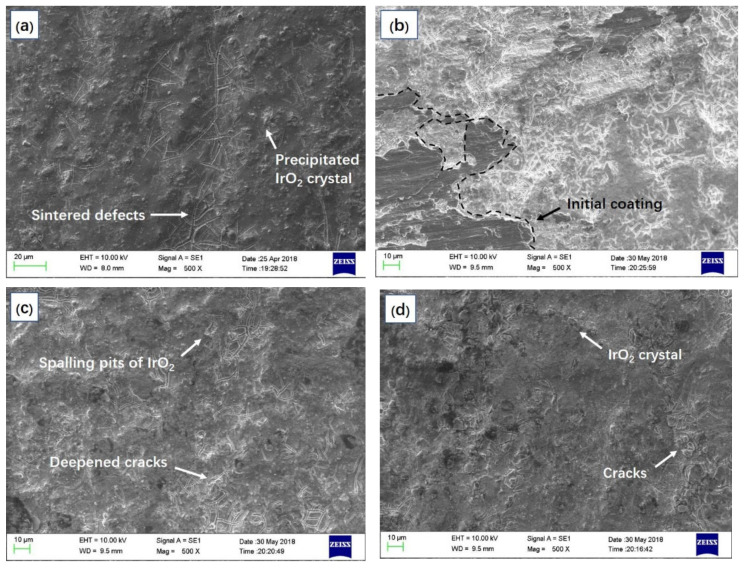
The SEM images of the primary anode surface after the simulated ICCP test: (**a**) raw anode metal, (**b**) anode metal from the cell with NA, (**c**) anode metal from the cell with 15 vol% of LFA, (**d**) anode metal from the cell with 25 vol% of LFA, and (**e**) anode metal from the cell with 40 vol% of LFA.

**Figure 6 materials-15-01977-f006:**
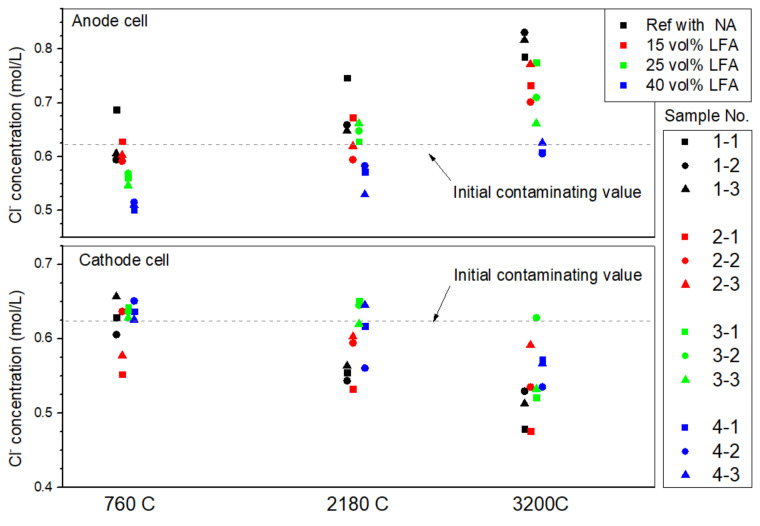
The measured chloride concentration of all simulated ICCP testing samples in different operation stages.

**Figure 7 materials-15-01977-f007:**
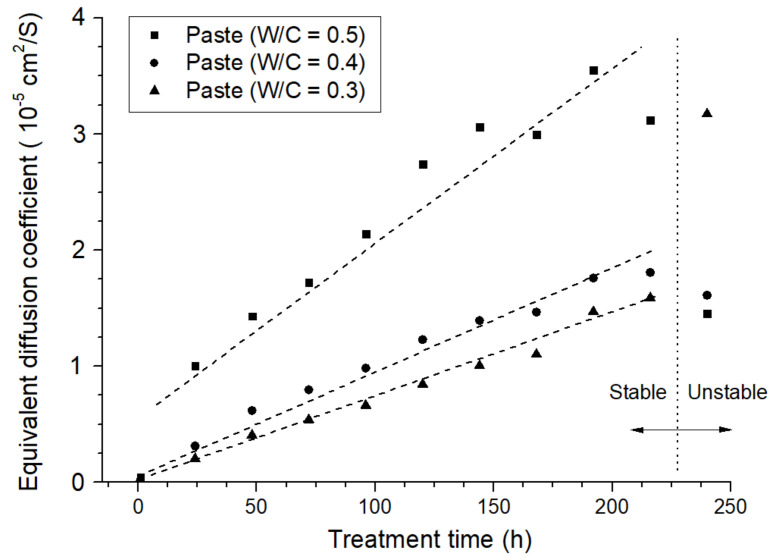
The time-dependent variation in the equivalent diffusion coefficient of the anode samples prepared with neat cement paste.

**Figure 8 materials-15-01977-f008:**
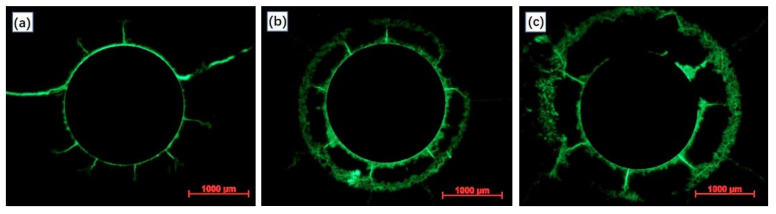
The fluorescence micrograph of anode samples prepared with neat cement paste in varied water to cement ratio after the constant potential polarization test: (**a**) W/C = 0.3, (**b**) W/C = 0.4, and (**c**) W/C = 0.5.

**Figure 9 materials-15-01977-f009:**
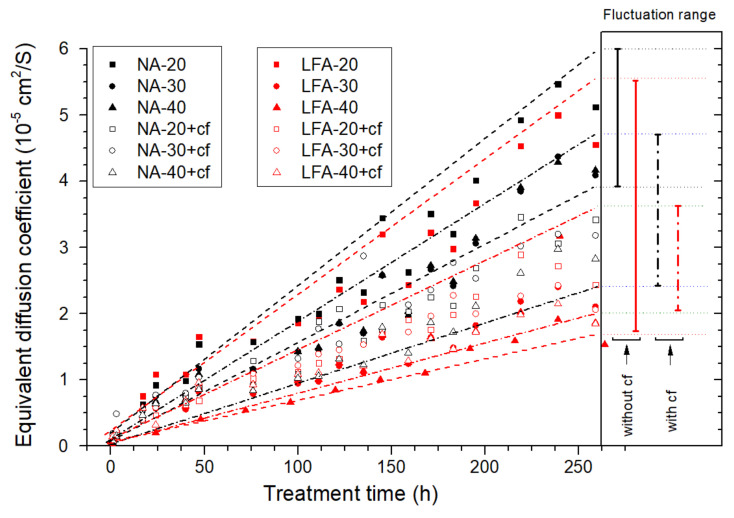
The time-dependent variation in the calculated equivalent diffusion coefficient of the anode samples prepared with different mixture proportions of secondary anode mortars.

**Figure 10 materials-15-01977-f010:**
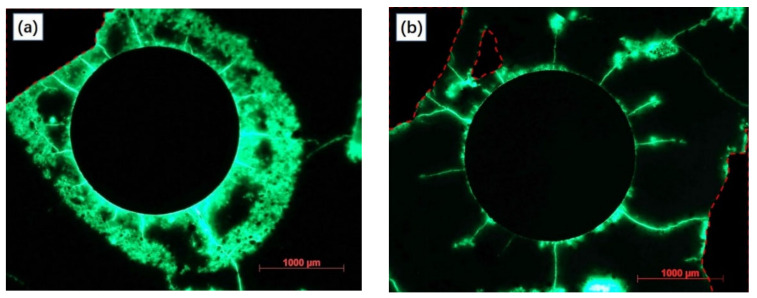
The fluorescence micrograph of anode samples with the maximum (**a**) or minimum (**b**) average growth rate of the equivalent diffusion coefficient.

**Figure 11 materials-15-01977-f011:**
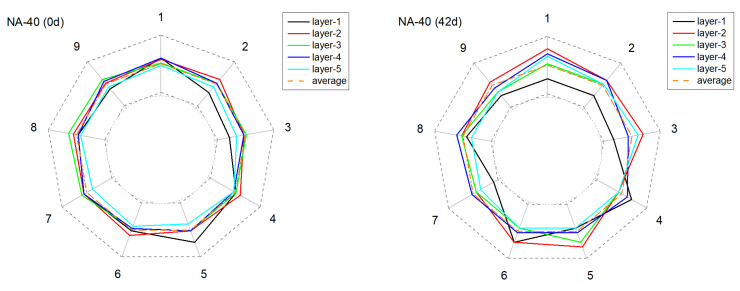
The deviation of the current from the mean value of current over time at different positions in the anode sample during the AAT (NA-40).

**Figure 12 materials-15-01977-f012:**
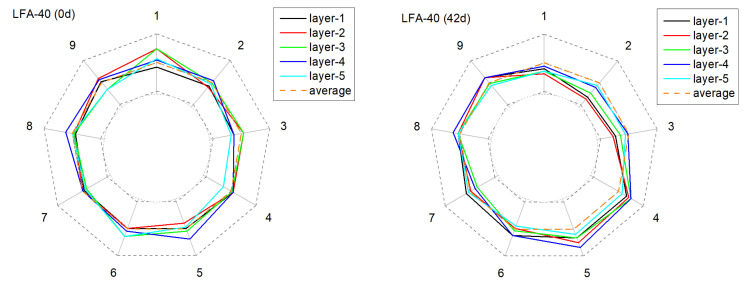
The deviation of the current from the mean value of current over time at different positions in the anode sample during the AAT (LFA-40).

**Figure 13 materials-15-01977-f013:**
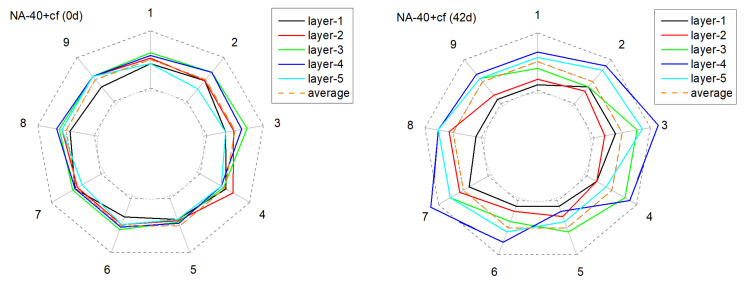
The deviation of the current from the mean value of current over time at different positions in the anode sample during the AAT (NA-40+cf).

**Figure 14 materials-15-01977-f014:**
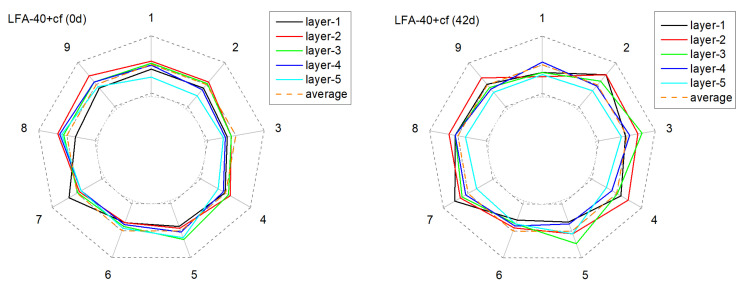
The deviation of the current from the mean value of current over time at different positions in the anode sample during the AAT (LFA-40+cf).

**Figure 15 materials-15-01977-f015:**
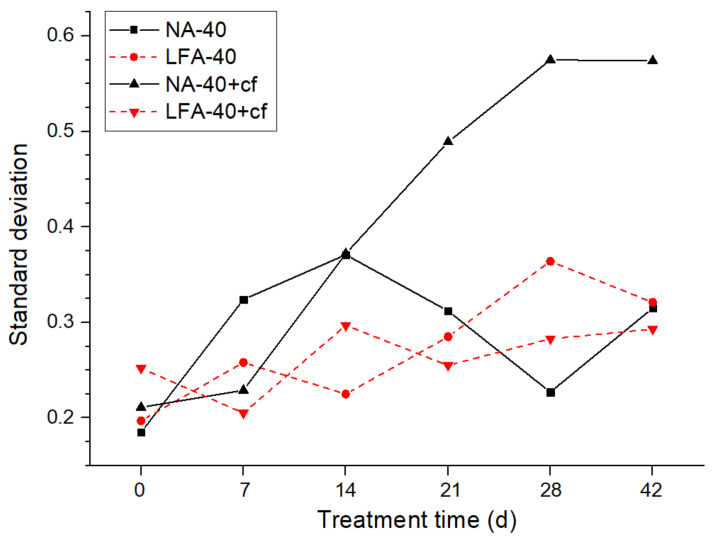
The relation between the normalized standard deviation of the detected current and the AAT treatment time.

**Table 1 materials-15-01977-t001:** Chemical composition of PII 42.5 Portland cement (wt%).

	SiO_2_	Al_2_O_3_	Fe_2_O_3_	CaO	MgO	K_2_O	Na_2_O	SO_3_	rest	LOI *
**Cement**	21.60	4.35	2.95	63.81	1.76	0.51	0.16	2.06	1.61	1.19
**Fly ash**	49.82	26.74	6.66	10.39	1.24	1.43	0.95	1.07	1.70	−0.70
**Slag**	31.63	17.98	0.42	38.10	7.55	0.46	0.40	2.56	0.90	0.00

* LOI, loss on ignition.

**Table 2 materials-15-01977-t002:** The mixture proportions of concrete (kg/m^3^).

Water	Cement	Slag	Fly Ash	Fine Aggregates	Coarse Aggregates	Super Plasticizer
173.8	246.0	70.0	35.0	741.0	1631.0	2.3

**Table 3 materials-15-01977-t003:** Aggregate configuration scheme of anode cells of the simulated ICCP system.

No.	NA/cm^3^	NA/g	LFA/cm^3^	LFA/g
1	26.0	67.6	0.0	0.0
2	16.0	41.6	10.0	10.0
3	10.0	26.0	16.0	16.0
4	0.0	0.0	26.0	26.0

**Table 4 materials-15-01977-t004:** Information of anode sample mix proportions.

Sample Type	Neat Paste	Mortar
W/C ratio	0.3, 0.4, 0.5	0.4
Carbon fiber	0.0 vol%	0.75 vol%
Aggregates	0.0 vol%	NA or LFA (20.0, 30.0, 40.0 vol%)

**Table 5 materials-15-01977-t005:** The average growth rate of the equivalent diffusion coefficient and the fitted RSS of different anode samples.

No.	DA Slope (cm^2^·s^−1^·h^−1^)	RSS
NA-20	1.96 × 10^−7^	2.25 × 10^−10^
NA-30	1.53 × 10^−7^	1.81 × 10^−10^
NA-40	1.57 × 10^−7^	1.43 × 10^−10^
LFA-20	1.70 × 10^−7^	2.18 × 10^−10^
LFA-30	0.81 × 10^−7^	0.51 × 10^−10^
LFA-40	0.69 × 10^−7^	0.48 × 10^−10^
NA-20+cf	1.24 × 10^−7^	0.93 × 10^−10^
NA-30+cf	1.17 × 10^−7^	1.40 × 10^−10^
NA-40+cf	1.01 × 10^−7^	0.78 × 10^−10^
LFA-20+cf	1.02 × 10^−7^	0.47 × 10^−10^
LFA-30+cf	0.85 × 10^−7^	0.53 × 10^−10^
LFA-40+cf	0.69 × 10^−7^	0.51 × 10^−10^

## Data Availability

The data presented in this study are available on request from the corresponding author.

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
