# Peer review of "The Effect of Lightweight Functional Aggregates on the Mitigation of Anode Degradation of Impressed Current Cathodic Protection for Reinforced Concrete"

_materials, 2022, doi:10.3390/ma15051977_

Round 1

Reviewer 1 Report

The impact of the study to real applications is not reported. The whole study will stay in theory and simulation. What are the practical findings of this study to concrete applications?

Figure 5 has five images. What do they mean. Label and explain the content of each image with differences. Use arrows.

Figure 11, 12, 13, and 14 have similarities. Minimize the same level of figures.

The study should clearly and explicitly report the followings. Figure 1 is clear. However, the article gets messy when you read further.

1) What are the simulated studies?

2) What are the experimental studies?

3) What are the differences between the findings of simulated and experimental studies? What are the error factors.

4) The studies did not consider the factor of interaction of the parameters. Explain. Why?

5) Your abstract indicates that you did not perform optimization studies. But, your conclusion indicates that the potential adsorption capability of impregnated modified agar gel within LFA could be the most reasonable explanation for this optimization mechanism.

Delete the numbers 1, 2, 3, 4 in Conclusions.

Author Response

  1. The impact of the study to real applications is not reported. The whole study will stay in theory and simulation. What are the practical findings of this study to concrete applications?

——Thanks for the comments, the authors would like to explain this comment as follows: the advantage of conventional test methods was that they can effectively characterize the polarization stability and accelerated service life of anode samples. Despite knowing the primary reason for a compromised anode performance was the propagation of anode acidification, no corresponding methods can be quantitatively utilized to relate the degree of local acidification and the degradation of anode performance. In this study, the simulated ICCP testing apparatus was designed based on parameters from ICCP engineering applications, therefore, the test results can be maximally related to the actual situation. Moreover, the design of anode cells can establish a quantitative relationship between anode performance and local acidification degree, which is critical for the emphasizing of ICCP treatment optimization via local acidification inhibition. In addition, despite the conventional investigations have discovered the nonuniform current distribution of ICCP system, they failed to quantitatively characterize the relationship between current distribution and anode degradation, nor to mention the optimization effects of anode modification on the conductivity uniformity. However, with the help of the AAT-combined conductivity uniformity tests scheme, the alteration of conductivity uniformity of anode samples can be obtained.

  1. Figure 5 has five images. What do they mean. Label and explain the content of each image with differences. Use arrows.

——Thanks for the comments, the five SEM images were presented to illustrate the differences of MMO coating integrity of primary anode wire in varied anode conditions after the same accelerated polarizing scheme. The purpose of listing these images was to demonstrate the relationship between local acidification and the degree of degradation of primary anode wire. The Arrows have been added to the images in the revised manuscript.

  1. Figure 11, 12, 13, and 14 have similarities. Minimize the same level of figures.

——Thanks for the comments, images corresponded to 14 d`s and 28 d`s AAT were removed, the level of figures has been minimized in the revised manuscript. Despite the intermediate process were removed, quantitative characterization on the alteration of conductivity uniformity can be traced in presented data from Fig.15.

4. The study should clearly and explicitly report the followings. Figure 1 is clear. However, the article gets messy when you read further.

1) What are the simulated studies?

2) What are the experimental studies?

3) What are the differences between the findings of simulated and experimental studies? What are the error factors.

4) The studies did not consider the factor of interaction of the parameters. Explain. Why?

5) Your abstract indicates that you did not perform optimization studies. But, your conclusion indicates that the potential adsorption capability of impregnated modified agar gel within LFA could be the most reasonable explanation for this optimization mechanism.

——Thanks for the comments, the authors would like to explain the detailed comments as follows:

1)This study did not contain pure simulation, the so-called simulated ICCP testing was an empirical study based on test apparatus as presented in Fig.1. In fact, the testing apparatus was designed based on parameters from engineering applications, detailed information were presented in section 2.2.1. To be specific, this test was named with “simulated” because of two cells filled with simulated concrete pore solution were introduced to make local acidification monitoring possible, and the designed two cells were different from real ICCP treatment system.

2)The answer is the same with the previous question.

3)In this study, the designed simulated ICCP testing apparatus was beneficial for a real-time monitoring of local acidification at the anode site. In addition, the design also achieved the real-time sampling and monitoring of chloride withdraw effects during an ICCP treatment (as presented in Fig.6). Again, this study only contains experimental results.

4)This study and the designed simulated ICCP testing apparatus focused on the relationship between local acidification and related performance alterations through experimental means.

5)Thanks for this comment, based on the obtained results, as presented in Fig.6, it is clear that the presence of LFA was related to a decreased chloride concentration within anode cells. Since the aggregate type and volume were the only differences between paralleled samples, the adsorption capability of agar-gel within LFA particles was the most likely explanation for the decreasing of chloride ions. The authors have modified the description in the revised manuscript.

5. Delete the numbers 1, 2, 3, 4 in Conclusions.

——Thanks for the comments. The authors have deleted the numbers in the revised manuscript.

Reviewer 2 Report

Dear Authors,

Thank you for your excellent written manuscript, just few remarks to the manuscript:

  1. Some additional information on your materials characterization (section 2.1) would be appreciated.
  2. Please add some description and marking directly to your SEM images to see the difference clearly among them.
  3. Please elaborate information on the novelty of the study in your conclusions.

Author Response

  1. Some additional information on your materials characterization (section 2.1) would be appreciated.

——Thanks for the comments. The detailed information on applied materials has been added in the revised manuscript.

  1. Please add some description and marking directly to your SEM images to see the difference clearly among them.

——Thanks for the comments, markings on typical morphology have been added on the SEM images in the revised manuscript.

  1. Please elaborate information on the novelty of the study in your conclusions.

 ——Thanks for the comments, the first paragraph in this section has been rewritten and the conclusions have been modified in the revised manuscript.

Round 2

Reviewer 1 Report

This is a very complicated and hard-to-grasp paper. I am happy with the additions made by the authors. However, I am not the subject matter expert to fully judge the quality of the paper.